# *Zuccagnia punctata* Cav. Essential Oil into Poly(ε-caprolactone) Matrices as a Sustainable and Environmentally Friendly Strategy Biorepellent against *Triatoma infestans* (Klug) (Hemiptera, Reduviidae)

**DOI:** 10.3390/molecules26134056

**Published:** 2021-07-02

**Authors:** Sandra López, Alejandro Tapia, Julio Zygadlo, Raúl Stariolo, Gustavo A. Abraham, Pablo R. Cortez Tornello

**Affiliations:** 1Instituto de Biotecnología-Instituto de CienciasBásicas, Universidad Nacional de San Juan, Av. Libertador General San Martín 1109 (O), San Juan 5400, Argentina; slopez@unsj.edu.ar; 2Instituto Multidisciplinario de Biología Vegetal, Cátedra de Química Orgánica, Facultad de Ciencias Exactas, Físicas y Naturales, Universidad Nacional de Córdoba, Córdoba 5016, Argentina; jzygadlo@unc.edu.ar; 3Coordinación Nacional de Control de Vectores, Córdoba 5000, Argentina; stariolo@yahoo.com.ar; 4Instituto de Investigaciones en Ciencia y Tecnología de Materiales, INTEMA (UNMdP-CONICET), Mar del Plata 7600, Argentina; gabraham@fi.mdp.edu.ar

**Keywords:** Chagas’ disease transmission, triatomines, peridomiciliary use, Argentina

## Abstract

The main strategies against *Triatoma infestans* (primary vector responsible for the Chagas disease transmission) are the elimination or reduction of its abundance in homes through the application of insecticides or repellents with residual power, and environmental management through the improvement of housing. The use of plant-derived compounds as a source of therapeutic agents (i.e., essential oils from aromatic plants and their components) is a valuable alternative to conventional insecticides and repellents. Essential oil-based insect repellents are environmentally friendly and provide reliable personal protection against the bites of mosquitoes and other blood-sucking insects. This study investigates, for the first time to our knowledge, the potential repellent activity of *Zuccagnia punctata* essential oil (ZEO) and poly(ε-caprolactone) matrices loaded with ZEO (ZEOP) prepared by solvent casting. The analysis of its essential oil from aerial parts by GC–FID and GC-MS, MS allowed the identification of 25 constituents representing 99.5% of the composition. The main components of the oil were identified as (−)-5,6-dehydrocamphor (62.4%), alpha-pinene (9.1%), thuja-2, 4 (10)-diene (4.6%) and dihydroeugenol (4.5%). ZEOP matrices were homogeneous and opaque, with thickness of 800 ± 140 µm and encapsulation efficiency values above 98%. ZEO and ZEOP at the lowest dose (0.5% wt./wt., 96 h) showed a repellency of 33 and 73% respectively, while at the highest dose (1% wt./wt., 96 h) exhibited a repellent activity of 40 and 66 %, respectively. On the other hand, until 72 h, ZEO showed a strong repellent activity against *T. infestans* (88% repellency average; Class V) to both concentrations, compared with positive control N-N diethyl-3-methylbenzamide (DEET). The essential oils from the Andean flora have shown an excellent repellent activity, highlighting the repellent activity of *Zuccagnia punctata*. The effectiveness of ZEO was extended by its incorporation in polymeric systems and could have a potential home or peridomiciliary use, which might help prevent, or at least reduce, Chagas’ disease transmission.

## 1. Introduction

Chagas disease is an extraordinarily complex zoonosis that is present throughout the territory of South America, Central America, and Mexico, and continues to represent a serious threat to the health of the countries of the region. This illness affects 6 to 7 million people around the world [1]. The main strategies used to interrupt vector transmission of *Trypanosoma cruzi* by *Triatoma infestans* (primary responsible for the Chagas disease transmission) are the elimination or reduction of its abundance in homes through the application of insecticides with residual power, and environmental management the improvement of housing [1]. 

Pyrethroid insecticides have been used for over 20 years to control the vectors of Chagas disease in Argentina and other Latin American countries [2]. From them, deltamethrin has been intensively used for the chemical control of *T. infestans* “vinchucas” and in general has displayed an effect as a highly effective triatomicide [3]. In 2002, the health authorities for the control of vectors in Argentina reported failures in the chemical control of *T. infestans* associated with different levels of resistance to pyrethroids [4,5,6]. This fact could be caused by the rapid degradation of the active compound, as shown in several studies [7]. 

To overcome these problems, active agents have been incorporated in polymeric systems allowing their protection and sustained release. Polymer-based systems have shown a longer lasting effect than traditional suspension concentrate formulations, both under experimental and field conditions [8]. Nanofibrous mats containing citriodiol as biorepellent against *Aedes aegypti* mosquitoes and its incorporation into a layered fabric were recently studied [9]. Monolithic and core-enriched nanofibrous mats with repellent activity were successfully obtained, and core-enriched mats displayed a 100% of repellency for 34 days. Moreover, compounds of botanical origin such as the essential oils from aromatic plants and their components provide an alternative to conventional insecticides and repellents. Essential oil-based insect repellents are environmentally friendly and provide dependable personal protection against the bites of mosquitoes and other blood-sucking insects [10]. On the other hand, the essential oils have the advantage of being agents of low toxicity in mammals, little residual life in the environment, and fewer requirements imposed by the legal framework, because they enjoy social acceptance due to the widespread use of aromatic species [11]. 

The essential oils from the Andean flora growing in the province of San Juan located in the center-west of the Argentine have shown an excellent repellent activity repellents to *Triatoma infestans* (Klug) (Hemiptera, Reduviidae), the vector of Chagas disease, since they constitute a rich source of bioactive compounds that are biodegradable into nontoxic products [12,13,14], effect that could be enhanced by incorporating them into polymeric systems, which are considered a suitable strategy for time and distribution-controlled repellent delivery [13]. The use of repellents against *T. infestans* vectors might help prevent, or at least reduce, Chagas’ disease transmission. The resinous species including the genus *Larrea* in Argentina (*Larrea ameghinoi*, *L. cuneifolia*, *L. divaricata,* and *L. nitida*), vernacular name “jarillas” and *Zuccagnia punctata*, commonly called “jarilla macho” are used extensively in traditional medicine in Argentina Andean communities for the treatment of injuries and bruises, and a good disinfectant of wounds, repellent of insects, for roof construction in rural areas and as a vegetable fuel for cooking food. [15,16].

On the other hand, poly(ε-caprolactone) (PCL) is a well-known aliphatic biocompatible polyester with a glass transition temperature at −60 °C and melting temperature between 59–64 °C. Its semicrystalline structure and hydrophobic character allow PCL to exhibit a long degradation time under humidity or physiological conditions of around 2 years. This is an attractive property for long-term applications in bioactive agent delivery [17,18,19,20]. There are reports from Peres et al. in which they propose the encapsulation of essential oil of fruit and leaves of *Xylopia aromatica* in PCL nanoparticles [21]. The nanoencapsulation of these bioactive compounds promotes their protection from environmental degradation and prolongs their biological activity. De Ávila et al. reported the preparation of PCL microparticles with encapsulated citronella oil through an emulsion technique followed by solvent evaporation [22]. Akolade et al. reported the microencapsulation of eucalyptol in poly(ethylene glycol) and PCL using particles from gas-saturated solutions [23]. Unalan et al. reported the fabrication and characterization of various concentrations of peppermint essential oil (PEP) loaded on PCL electrospun fiber mats for wound healing applications, where PEP was intended to impart antibacterial activity to the fibers [24].

This study investigated for the first time the potential repellent activity of *Zuccagnia punctata* essential oil and its incorporation in PCL matrices for increasing the duration of the repellent activity.

## 2. Results and Discussion

### 2.1. Essential Oil Composition, Yield, and Spectroscopy Characterization

The essential oil yield was 0.25% (v/wt.); δ^25^: 0.96 g/mL. Regarding the chemical profile, a total of 25 compounds amounting 99.5% of the oil were identified according [25,26,27]. The main constituents are showed in Table 1 and include (−)-5,6-dehydrocamphor (62.4%), alpha-pinene (9.1%), thuja-2,4(10)-diene (4.6%), terpinen-4-ol (4.4%), verbenone (3.1%) and dihydroeugenol (4.5%). The monoterpenes represented the main portion of the oil accounting for 89.2% with a high percentage of oxygenated monoterpenes (69.9%). Hydrocarbon sesquiterpenes accounted for 4.7 %. Among them, the most abundant was epi-beta-santalene (2.1%). Epi-alpha-cadinol was the oxygenated sesquiterpenoid detected (0.7%). 

Constituents listed in order of increasing retention indices (RI). Unidentified components less than 0.1% are not reported. Temperature-programmed RI referred to n-alkanes, determined on a HP-5MS capillary column. Percentage values less than 0.1% are denoted as t (traces). Method of identification of minor constituents: 1corresponds to comparison of GC-MS data and RI with those of the volatile oil ADAMS, Wiley and NBS computer mass libraries, 2corresponds to comparison of GC-MS data and RI with those of authentic samples.

The chemical composition as well as the antifungal activity of the *Zuccagnia punctata* essential oil collected in the province of San Juan have been previously reported [27], standing out the presence of (−)-5,6-dehydrocamphor (56.5%), linalool (14.5%) and cis-linalool oxide THF (3.4%). The chemical composition of the essential oil reported here shows also that the main component is (−)-5,6-dehydrocamphor (62.4%), with some differences in minor components. The chemical composition is genetically determined (intrinsic factors) and on the other hand, environmental conditions (extrinsic factors) may be responsible for significant variations in the chemical composition of plants [28]. Essential oils can qualitatively and quantitatively change their chemical composition due to climatic factors, the composition of the soil, the plant organ, age, seasonality, and the phase of the circadian cycle [29,30,31].

### 2.2. Morphological Characterization

The Figure 1 shows a disc of 10 mm cut from the ZEOP matrix and the SEM image of ZEOP 1% sample. ZEOP matrices were homogeneous and opaque with thickness of 800 ± 140 µm, as measured with a low force caliper. The SEM micrograph exhibited a characteristic morphology of PCL matrices prepared by solvent casting. The surface porosity could favor the evaporation of essential oils from the polymer matrix regions with dispersed oil.

### 2.3. Thermal Properties and Crystallinity

The Figure 2 shows the DSC thermograms. PCL pellets exhibited a characteristic thermogram with a melting temperature of 65.6 °C and crystallinity of 59.3%. PCL matrices showed a decrease in the melting point (Tm = 62 °C) and slight increase in crystallinity (63.8%) which can be attributed to the matrix formation during solvent casting. The incorporation of ZEO led to a decrease in Tm values with the increase in the oil content (ZEOP 0.5%, 61.4 °C and ZEOP 1%, 59.0 °C). This phenomenon agrees with the decrease in the crystallinity degree (ZEOP 0.5%, 61%, and ZEOP 1%, 59.6%), and it could be ascribed to the incorporation of oil, which makes difficult the crystallization process during the solvent evaporation. The ZEO thermogram did not show thermal events in the explored temperature by subheadings. It should provide a concise and precise description of the experimental results, their interpretation, as well as the experimental conclusions that can be drawn.

TGA curves of ZEO are shown in Figure 3. The results indicated that the thermal behavior of *Zuccagnia punctata* is simple and present only one thermal process. A continuous weight loss starting at 62.6 °C and continued until 358.2 °C. These values indicate that ZEO decomposition with temperature begins above room temperature, and therefore it is stable at the temperature of use of ZEOP matrices.

### 2.4. ZEO Loading Capacity and Encapsulation Efficiency

Table 2 shows the ZEO content in ZEOP matrices per mass unit of sample (Mc), loading capacity (LC), and encapsulation efficiency (EE). LC values of ZEOP 0.5% (0.48%), and ZEOP 1% (0.97%) matrices showed that ZEO content were slightly lower than the amount present in the polymeric solution. On the other hand, EE values showed encapsulation of ZEO above 98%. These values were consistent with those reported by Peres et al. [21] for PCL nanoparticles containing the essential oils of *Xylopia* (95%), and higher than the obtained values for PCL microparticles loaded eucalyptol (77 %) [23] and PCL fibers with peppermint oil (37 %) [24]. These results indicate that PCL is a suitable polymer for ZEO encapsulation.

### 2.5. Repellent Activity against Triatoma Infestans Nymphs

The results of the assay of repellency of the ZEO and ZEOP are shown in Table 3 and Table 4.

The ZEO showed excellent repellent properties on *T. infestans* between 1 and 72 h, for the two concentrations of the oil tested (Table 3 and Table 4). The percentage of repellence did not change significantly with time (no effect within subjects, *p* > 0.05), and no significant relationship was observed between time points and oil treatment (*p* > 0.05). The essential oil was Class V, which is the one with the highest repellency according to the methodology used; the mean values obtained were 88.3 and 88.8%, for concentrations of 0.5 and 1% (wt./wt.), respectively. For both concentrations, the repellent activity decays between 72 and 96 h until a Class II repellent activity. This short-term action regarding the duration of the effect may be what limits the use of insect repellent products based on essential oils, according to previous reports, it may be related to the rapid volatilization and short time of action [32]. On the other hand, significant differences in average percentage of repellence were observed between the ZEO treatment and blank control (effects between subjects, *p* < 0.05).

In a previous report, the chemical composition, anti-insect, and antimicrobial activity of *Baccharis darwinii* essential oil from Argentina, Patagonia were reported. The major components with recognized anti-insect and antimicrobial activity were identified, including limonene (47.1%), thymol (8.1%) and, 4-terpinelol (6.4%). The in vitro evaluation of the anti-insect properties showed promising insecticidal activity against *Ceratitis capitata* (LD50 19.9–31.0 g/fly for males and females respectively at 72 h) and repellent activity against *T. infestans* (average repellence 92%, Class V) [33]. The potential of the Andean medicinal flora of Argentina as a source of essential oils with repellent activity has been reported in the last decade, together with the chemical profile of volatile compounds [12,13,14].

Regarding ZEOP, the repellent activity showed a low activity during the first 24 h (Class III) and it was growing until 96 h (Class IV) to both concentrations assayed (Table 3 and Table 4). On the other hand, significant differences in average percentage of repellence were observed between the ZEOP treatment and blank control (effects between subjects, *p* < 0.05).

Nanoproducts developed using natural products have been highlighted as ecologically and economically sustainable alternatives for effective control of crop pest and other vectors of human incidence such as mosquitoes and triatomines. The strong activity of limonene and β-pinene against *Tribolium castaneum* has been informed; however, the high volatility and hydrophobicity hinder the use of these monoterpenes as a large-scale pest control agent [34].Recently, has been reported the nano-emulsification of monoterpenes and essential oil allowed their incorporation into an aqueous matrix without losing its repellent activities [34], which gives support to the results reported here. The controlled release systems for repellents comprise polymer micro/nanocapsules, micro/solid lipid nanoparticles, nanoemulsions/microemulsions, liposomes/niosomes, nanostructured hydrogels and cyclodextrins [35]. There are many formulations based on micro and nanocapsules containing DEET and essential oils to increase repellent action time duration and decrease permeation and consequently, systemic toxicity [36]. Limonene essential oil successfully encapsulated in microcapsules of chitosan showed a slow and prolonged liberation profile by volatilization [36]. The ZEOP has also shown a slow and prolonged release during 96 h.

The oil from *Z. punctata*, one of the endemic resinous species in Argentina that is extensively used in the traditional medicine of Argentina and Andean people for various purposes, has shown significant potential as a biorepellent against the vector of Chagas disease. Repellent activity is prolonged significantly if the oil is supported in a polymeric system.

## 3. Materials and Methods

### 3.1. Chemicals 

All solvents used were of analytical grade. Chloroform was purchased from Fisher (Walham, MA, USA); acetone and methanol (MeOH) grade UHPLC from J.T. Baker (Phillipsburg, NJ, USA) and dichloromethane (DCM) from Aldrich Chemical Co. (St. Louis, MO, USA). Poly(ε-caprolactone) (PCL, Mw 80000 g/mol) and *N,N*-diethyl-3-methylbenzamide (DEET) were purchased from Aldrich Chemical Co. (USA). Ultra-pure water (<5 µg/L) was obtained from a purification system Arium 61316-RO plus Arium 611 UV (Sartorius, Göttingen, Germany). 

### 3.2. Plant Material

The aerial parts of *Zuccagnia punctata* Cav. (Fabaceae, Caesalpinoideae) were collected in January 2018, on Iglesia district, province of San Juan (Argentina) at an altitude of 1800 m above sea level. The species has been previously identified by Dr Gloria Barboza, IMBIV (Instituto Multidisciplinario de Biología Vegetal, Facultad de Ciencias Exactas, Físicas y Naturales, Universidad Nacional de Córdoba, Argentina). A voucher specimen has been previously deposited at the herbarium of the Botanic Museum of Córdoba (CORD 1125).

### 3.3. Essential Oil Extraction and Chemical Analysis

Fresh aerial parts (500 g) were subjected to hydrodistillation for 2 h using a Clevenger type apparatus. The yields were averaged over two experiments and calculated according to dry weight of plant material. Essential oils (ZEO) were stored at −1 °C in airtight micro-tubes prior to chemical analysis. Qualitative data were determined by GC–FID and GC-MS. Gas chromatography-mass spectrometry analyses were carried out on a Hewlett-Packard 5890 II gas chromatograph coupled to a Hewlett-Packard 5989 B mass spectrometer, using a methyl silicone HP-5MS (crosslinked 5% PH ME Siloxane) capillary column (30 m × 0.25 mm), film thickness 0.25 µm. Samples were analyzed using the following GC-MS conditions: oven temperature program: 50–250 °C at 3 °C/min, carrier gas: helium, 1.5 mL/min; injection temperature: 250 °C, FID detector temperature: 300 °C; split mode ratio of 1:60. Additional parameters in the mass spectrometer unit: ion source temperature of 250 °C; ionizing voltage of 70 eV; scan range from *m/z* 35 to *m/z* 300. The identification of components was performed with the use of the volatile oil ADAMS library together with retention indices of reference compounds and built-in Wiley and NBS peak matching library search systems. Quantitative percentage composition was determined from the GC peak areas without correction factors [25,26,27].

### 3.4. Preparation of Zuccagnia punctata Essential Oil Loaded Polymeric Systems

PCL solutions of 10 wt./v % were prepared by dissolving PCL pellets in a 5 mL of DCM:MeOH solvent mixture (50:50 by volume) under magnetic stirring. For the preparation of polymeric matrices of PCL containing *Zuccagnia punctata* essential oil (ZEOP), 0.5 and 1 % wt./wt. of ZEO with respect to PCL were added to the solution. The selected solvent mixture allowed the complete dissolution of ZEO and PCL.

ZEOP were prepared by solution casting onto a Petri dish (4.6 mm in diameter) and dried in a fume hood at room temperature for 24 h. Samples were subsequently vacuum dried to remove residual solvent. Disc samples of 10 mm were cut and stored at room temperature under vacuum until use.

### 3.5. PCL, ZEO, and ZEOP Matrices Characterization

The morphology of matrices was examined by scanning electron microscopy (SEM, JEOL JSM6460 LV, Peabody, MA, USA) operated at 15 kV. Samples were sputter-coated with gold during 15 min in a chamber evacuated to 500 mTorr (Sputter coater, Desk II, Denton Vacuum, Moorestown, NJ, USA). Thermal properties of PCL pellets, ZEO, ZEOP and PCL matrices were determined by differential scanning calorimetry (DSC, TA instrument, Model Q-2000, New Castle, DE, USA). Scans were carried out at a heating rate of 10 °C/min. Glass transition temperature was taken as the onset of the transition. The degree of crystallinity of PCL (Xc) was calculated as:Xc (%) = (ΔHm experimental/ΔHmtheoretical) × 100(1)
where the theoretical melting heat (ΔHm) for pure high molecular weight PCL was taken as 148.05 J/g [37]. Thermogravimetric Analysis (TGA) was conducted to study the thermal stability of ZEO. TGA data were obtained using a thermogravimetric analyzer (TA instrument, Model Q-500, New Castle, DE, USA). A sample of 5–10 mg was accurately weighed in an aluminum pan and the measurement was conducted at heating rate of 10 °C/min under nitrogen purging.

*Zuccagnia punctata* essential oil content was determined by ultraviolet-visible spectroscopy using an Agilent 8453 spectrometer (Santa Clara, CA, USA) equipped with a diode array system. A predetermined amount of sample was dissolved in DCM:MeOH (1:1 by volume), and quantification was carried out observing the absorption band at λ = 280 nm. At least three measurements were performed.

The loading capacity (LC) was calculated from the ratio between the ZEO mass in the sample (mZ) and the polymer mass (mP) in ZEOP matrix.
LC (%) = (mZ)/(mP) × 100(2)

The encapsulation efficiency (EE) was calculated as:EE (%) = (mZf/mPCLf)/(mZi/mPCLi) × 100(3)
where mZfis the mass of ZEO encapsulated, mZi the initial mass of ZEO, and mPCLf and mPCLi correspond to the final and initial mass of PCL, respectively.

### 3.6. Repellent Activity against Triatoma infestans Nymphs Fifth Instars

The bioassays were carried out according to [13,14]. *Triatoma infestans* nymphs fifth instar were provided by Servicio Nacional de Chagas (Córdoba, Argentina) and were used one day after receipt. 

Filter paper discs (9 cm in diameter) divided by halves were used. One half was treated with 0.5 mL of acetone solutions of the essential oils (0.5% and 1% wt./wt.) while the other half remained untreated. As control, circular white filter papers divided in two halves, one treated with 0.5 mL of acetone and the other untreated, were used. After solvent evaporation, filter paper discs were placed covering the floor of a Petri dish. Five starved nymphs of *T. infestans* (fifth instar) were released in the center of each Petri dish and maintained under controlled conditions of temperature 24 ± 2 °C, 50 ± 5% RH and photoperiod of 16 h L/8h D. Experiments were performed by quintuplicate. The same procedure was carried out with the polymer discs containing the essential oil at the same concentrations (0.5% and 1% wt./wt.). Insect distribution was recorded at 1, 24, 72, and 96 h of treatment. *N,N*-diethyl-3-methylbenzamide (DEET) was used as positive control at 0.5% (wt./v) and acetone as blank control.

Data were transformed into repellency percentage (RP %) as:RP % = (Nc − 50) × 2(4)

Nc corresponds to the percentage of nymphs in the blank half.

Positive values show repellence while negative values show attraction. Mean values were categorized according to the following scale: Class 0 (>0.01 to <0.1), I (0.1 to 20), II (20.1 to 40); III (40.1 to 60); IV (60.1 to 80), V (80.1 to 100) according to Talukder et al. [38]. Data were analyzed by repeated measures ANOVA to determine the overall significance of the repellence means between the time points and the effect of oil treatment as a factor between subjects. Data were analyzed with the statistical software SPSS 15.0 (SPSS Inc.).

## 4. Conclusions

Essential oil from *Zuccagnia punctata* Cav. (Caesalpinieae) growing in the province of San Juan, located in the center-west of the Argentine, may be a potential alternative repellent to *T. infestans* (Klug) (Hemiptera, Reduviidae), the vector of Chagas disease. This oil biorepellent constitutes a rich source of sustainable, bioactive, and biodegradable compounds, especially a high content of oxygenated monoterpenes, such as (−)-5,6-dehydrocamphor.

Polymeric matrices of PCL loaded with different amounts of *Zuccagnia punctata* were prepared and characterized. Essential oil content on polymeric matrices showed encapsulation efficiencies higher than 98%, and it is thermally stable. The essential oils from the Andean flora have shown an excellent repellent activity, highlighting the repellent activity of the essential oil of the medicinal species *Zuccagnia punctata*. The effectiveness of ZEO was extended by its incorporation in polymeric systems and could have a potential home or peridomiciliary use, which might help prevent, or at least reduce, Chagas’ disease transmission.

## Figures and Tables

**Figure 1 molecules-26-04056-f001:**
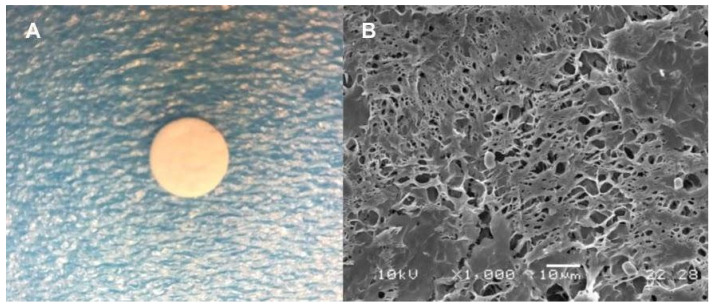
Morphological characterization, (**A**) Optical image of ZEOP 1% disc 10 mm, (**B**) SEM micrograph of ZEOP 1% (1000×).

**Figure 2 molecules-26-04056-f002:**
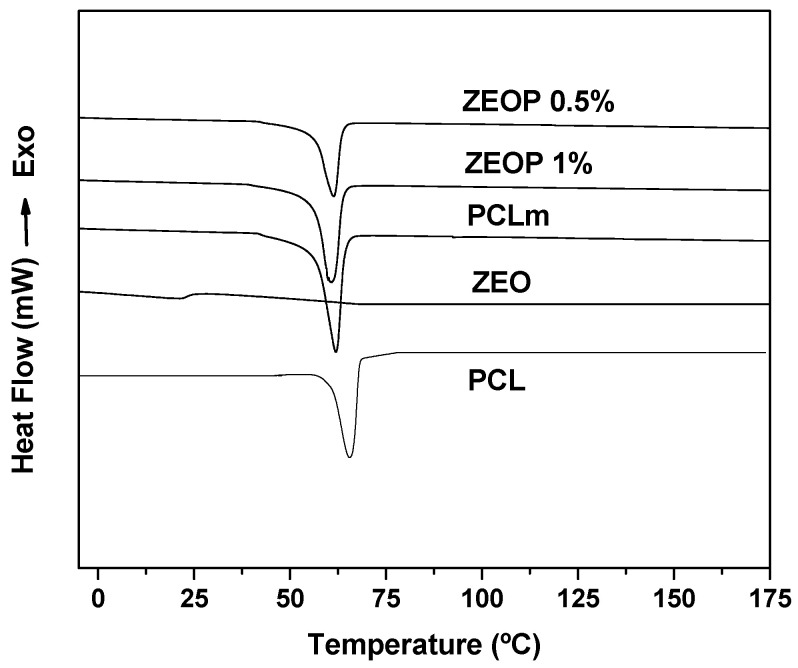
Differential scanning calorimetry thermograms corresponding to PCL, PCL matrix (PCLm), ZEO, ZEOP 0.5 % and ZEOP 1% formulations.

**Figure 3 molecules-26-04056-f003:**
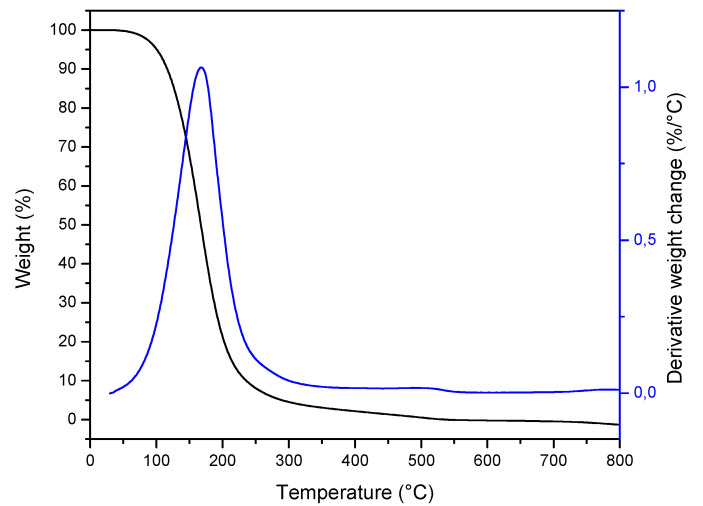
Thermal stability of ZEO: thermogravimetric and first-derivative of TGA curves.

**Table 1 molecules-26-04056-t001:** Chemical composition of the *Zuccagnia punctata* essential oil.

Peak	Component	RI	Area (%)	Identification Method
1	Alpha-thujene	928	t	1
2	Alpha-pinene	936	9.1	1, 2
3	Alpha-fenchene	950	0.3	1
4	Camphene	951	t	1, 2
5	Thuja-2,4(10)-diene	956	4.6	1
6	Myrcene	990	t	1, 2
7	Alpha-terpinene	1016	0.6	1
8	p-cymene	1025	t	1, 2
9	Limonene	1030	0.8	1, 2
10	Gamma-terpinene	1059	1.6	1
11	p-cymenene	1091	2.3	1
12	(−)-5,6-dehydrocamphor	1097	62.4	1
13	Terpinen-4-ol	1178	4.4	1, 2
14	Verbenone	1206	3.1	1
15	(E)-Cinnamyl alcohol	1304	0.4	1
16	Piperitenone	1343	t	1, 2
17	Eugenol (dihydro)	1369	4.5	1
18	(E)- caryophyllene	1419	0.8	1
19	Epi-beta-santalene	1447	2.1	1
20	Delta-amorphene	1512	0.6	1
21	(Z)-gamma-bisabolene	1515	0.1	1
22	Beta-curcumene	1516	0.2	1
23	Delta-cadinene	1523	0.4	1
24	(E)-gamma-bisabolene	1531	0.5	1
25	Epi-alpha-cadinol	1640	0.7	1
	Monoterpene hydrocarbons		19.3	
	Oxygenated monoterpenes		69.9	
	Phenylpropanoids		4.9	
	Sesquiterpenes hydrocarbons		4.7	
	Oxygenated sesquiterpenes		0.7	
Total			99.5	

**Table 2 molecules-26-04056-t002:** ZEO content incorporated in ZEOP matrices per mass unit of sample (Mc), loading capacity (LC), and encapsulation efficiency (EE).

Sample	Mc (±s.d.) (mg/g)	LC (±s.d.) (%)	EE (±s.d.) (%)
ZEOP 0.5%	4.87 ± 0.30	0.48 ± 0.08	98.45 ± 0.03
ZEOP 1%	9.75 ± 0.20	0.97 ± 0.10	98.51 ± 0.02

**Table 3 molecules-26-04056-t003:** Repellent activity of ZEO, and ZEOP against *T. infestans* nymphs fifth instars (mean ± SD, *n* = 5) at 0.5% (wt./wt.).

	Repellency (%) at 0.5 % (wt./wt.)			
	Treatments			
Time (h)	ZEO	ZEOP	Control ^3)^	DEET ^4)^
1	97.0 ± 2.0	33.0 ± 11.1	−12.0 ± 5.4	100.0 ± 0.0
24	92.0 ± 10.4	60.0± 24.0	−20.0 ± 32.0	100.0 ± 0.0
72	76.0 ± 5.3	60.0 ± 24.0	−100.0 ± 0.0	100.0 ± 0.0
Average repellency ^1)^	88.3 ± 5.0 ^a^	51.0 ± 8.1 ^a^	−44.0 ± 17.8 ^b^	100.0 ± 0.0 ^a^
Class ^2)^	V	III	-	V
96	33.0 ± 23.1 ^a^	73.0 ± 12.1 ^a^	−100.0 ± 0.0 ^b^	100.0 ± 0.0 ^a^
Class ^2)^	II	IV	-	V

^1)^ Average value of repellency in the three times. ^2)^ Repellence class according to scale: Class 0 (0.01 to 0.01%), Class I (0.1 to 20%), Class II (20.1 to 40%), Class III (40.1 to 60%), Class IV (60.1 to 80%), and Class V (80.1 to 100). ^a,b^ indicate significant difference at the 0.05 level according to Tukey test. ^3)^ Blank control acetone. ^4)^ Positive control DEET at 0.5% (wt./v).

**Table 4 molecules-26-04056-t004:** Repellent activity of ZEO, and ZEOP against *T. infestans* nymphs fifth instars, the vector of Chagas disease (mean ± SD, *n* = 5) at 1.0% (wt./wt.).

	Repellency (%) at 1 % (wt./wt.)			
	Treatments			
Time (h)	ZEO	ZEOP	Control ^3)^	DEET ^4)^
1	100.0 ± 0.0	46.6 ± 11.1	−12.0 ± 34.4	100.0 ± 0.0
24	93.0 ± 11.5	60.0 ± 6.0	−28.0 ± 32.0	100.0 ± 0.0
72	73.3 ± 23.1	66.0 ± 23.0	−100.0 ± 0.0	100.0 ± 0.0
Average repellency ^1)^	88.8 ± 5.0 ^a^	55.5 ± 7.0 ^a^	−46.7 ± 17.8 ^b^	100.0 ± 0.0 ^c^
Class ^2)^	V	III	-	V
96	40.0 ± 13.1	66.6 ± 6.7 ^a^	−100.0 ± 0.0 ^b^	100.0 ± 0.0
Class ^2)^	II	IV		V

^1)^ Average value of repellency in the three times. ^2)^ Repellence class according to scale: Class 0 (0.01 to 0.01%), Class I (0.1 to 20%), Class II (20.1 to 40%), Class III (40.1 to 60%), Class IV (60.1 to 80%), and Class V (80.1 to 100%). ^a,b,c^ indicate significant difference at the 0.05 level according to Tukey test. ^3)^ Blank control acetone. ^4)^ Positive control DEET at 0.5% (wt./v).

## Data Availability

Not applicable.

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
