# Peer review of "Zuccagnia punctata Cav. Essential Oil into Poly(ε-caprolactone) Matrices as a Sustainable and Environmentally Friendly Strategy Biorepellent against Triatoma infestans (Klug) (Hemiptera, Reduviidae)"

_molecules, 2021, doi:10.3390/molecules26134056_

Round 1

Reviewer 1 Report

The authors proposed to study « Zuccagnia punctata Cav. essential oil into poly(ε matrices)  as a sustainable and environ bio-repellent against Triatoma infestans Reduviidae».

The pape is well written. Minor change should be made before its acceptation.

Abstract : Background appears to times whereas methods are absent.

Results :

Line 199. Prestes et al. Reference number should be added.

Author Response

Reviewer 1

We are grateful to the Reviewer for their thoughtful and helpful comments on our manuscript. All their concerns have been addressed and changes in the text are highlighted in yellow. Please find below a point-by-point list of our answers 

Comments and Suggestions for Authors

The authors proposed to study «Zuccagnia punctata Cav. essential oil into poly(ε matrices) as a sustainable and environ bio-repellent against Triatoma infestans Reduviidae».

The paper is well written. Minor change should be made before its acceptation.

Abstract : Background appears to times whereas methods are absent.

Authors: We thank Reviewer #1 for her/his valuable comments. We agree with the Reviewer that Background subtitle was duplicated. As it is usual in this Journal, we have deleted subsections in Abstract section.

Results : Line 199. Prestes et al. Reference number should be added.

Authors: The reference for Peres et al (it is not Prestes) was added as [21].

Reviewer 2 Report

The manuscript entitled "Zuccagnia punctata Cav. essential oil into poly(ε-caprolactone) matrices as a sustainable and environ bio-repellent against Triatoma infestans Reduviidae)" has interesting subject and applicable findings. Also, it has acceptable English written and clarity. However, the following errors should be improved before its publication:

Line 27: Background or material and methods?

Line 29: Add space after "Results". Check manuscript for such mistyping (see for example lines 57 and 60).

Line 47: Add a reference for this sentence.

Lines 49-53: "through" was used two times in this sentence. Use another word.

Line 61: Add relevant reference(s).

Lines 62-65: Add relevant reference(s).

Lines 70-73: Attach this paragraph to the previous one.

Lines 74-106: Write them in a paragraph (same subject: repellent effects of agents containing botanicals).

Line 107: Repellence or repellent?

Line 137: What are these differences usually caused by? Explain more (with references).

Table 2: be consistent. Write the units in same format. Mg/g or mg.g-1? Consider it throughout the text.

Line 218-220: Add relevant reference for classification.

Line 221: According to Tukey test (p < 0.05). There is not any different letter. Add the results of mean comparison on all data.

Lines 245-247: This is an incomplete sentence. Rewrite it.

Lines 250-252: Add the relevant reference.

Line 255: Add a reference before comma.

Line 271: Add geographical location details.

Author Response

Reviewer 2

We are grateful to the Reviewer for their thoughtful and helpful comments on our manuscript. All their concerns have been addressed and changes in the text are highlighted in yellow. Please find below a point-by-point list of our answers 

Comments and Suggestions for Authors

The manuscript entitled "Zuccagnia punctata Cav. essential oil into poly(ε-caprolactone) matrices as a sustainable and environ bio-repellent against Triatoma infestans Reduviidae)" has interesting subject and applicable findings. Also, it has acceptable English written and clarity. However, the following errors should be improved before its publication:

Line 27: Background or material and methods?

Authors: We thank Reviewer #2 for her/his valuable comments. We agree with the Reviewer and a short mention was added:

“This study investigates, for the first time to our knowledge, the potential repellent activity of Zuccagnia punctata essential oil (ZEO) and poly(ε-caprolactone) matrices loaded with ZEO (ZEOP) prepared by solvent casting.”

As it is usual in this Journal, we have deleted subsections in Abstract section.

Line 29: Add space after "Results". Check manuscript for such mistyping (see for example lines 57 and 60).

Authors: We agree with the Reviewer. As it is usual in this Journal, we have deleted subsections in Abstract section. In addition, the manuscript was fully checked and some errors were corrected.

Line 47: Add a reference for this sentence.

Authors: 

Although in Molecules Journals (Instructions for Authors, Back Matter section) the use of websites as references is allowed, we have deleted the URL from the text "(http://www.who.int/mediacentre/factsheets/fs340/es/)". The reference for this sentence is [1].

Lines 49-53: "through" was used two times in this sentence. Use another word.

Authors: Second “through” was replaced by “by”.

Line 61: Add relevant reference(s).

Authors: The following reference was added: [7] Cycon, M.; Piotrowska-Seget, Z. Pyrethroid-Degrading Microorganisms and Their Potential for the Bioremediation of Contaminated Soils: A Review. Front Microbiol. 2016, 7, 1463. 

Lines 62-65: Add relevant reference(s).

Authors: The following reference was added: [8] Chiriac, A. P.; Rusu, A. G.; Nita, L. E.; Chiriac, V. M.; Neamtu, I.; Sandu, A. Polymeric Carriers Designed for Encapsulation of Essential Oils with Biological Activity. Pharmaceutics 2021, 13, 631.

Lines 70-73: Attach this paragraph to the previous one.

Authors: Paragraph was attached to the previous one.

Lines 74-106: Write them in a paragraph (same subject: repellent effects of agents containing botanicals).

Authors: The text was rephrased according to same subject: different repellents against T. infestans, and essential oil-loaded PCL systems. 

Line 107: Repellence or repellent?

Authors: The word was replaced. The correct word is “repellent” activity.

Line 137: What are these differences usually caused by? Explain more (with references). 

Authors: The following additional paragraph with its respective references was included in the main text.

“The chemical composition is genetically determined (intrinsic factors) and on the other hand, environmental conditions (extrinsic factors) may be responsible for significant variations in the chemical composition of plants [28]. Essential oils can qualitatively and quantitatively change their chemical composition due to climatic factors, the composition of the soil, the plant organ, age, seasonality and the phase of the circadian cycle [29-31].”

Table 2: be consistent. Write the units in same format. Mg/g or mg.g-1? Consider it throughout the text.

Authors: We agree with the reviewer, mg/g, g/ml, wt/wt, g/mol and other notations were used. We have revised and uniformized the format throughout the entire text.

Line 218-220: Add relevant reference for classification.

Authors: The following reference that support the assay was included in section 4.2. Repellent activity against Triatoma infestans nymphs fifth instars: [38]: Talukder, F. A.; Howse, P. E. Laboratory evaluation of toxic and repellent properties of the pithraj tree, Aphanamixis polystachya Wall & Parker, against Sitophilusoryzae (L.). Int. J. Pest Manag. 1994, 40, 274–279.

Line 221: According to Tukey test (p < 0.05). There is not any different letter. Add the results of mean comparison on all data.

Authors: The presentation of the data was reviewed as well as the statistical treatment. Additionally, a column containing positive control (DEET) was added.

Lines 245-247: This is an incomplete sentence. Rewrite it.

Authors: We agree with the reviewer. We rewritten the sentence.

Lines 250-252: Add the relevant reference.

Authors: We agree with the reviewer. A reference was added.

Line 255: Add a reference before comma.

Authors: We agree with the reviewer. A reference was corrected.

Line 271: Add geographical location details.

Authors: Geographic location details were added 

Reviewer 3 Report

I provided my comments as follows.

General comments:

The manuscript deals with the development of a essential oil-based bio-repellent against Triatoma infestans.The poly (ε-caprolactone) matrix loaded with Zuccagnia punctata essential oil was well characterized. However, the manuscript has important flaws and must be improved before being considered for publication in Molecules.

The following are my considerations:

1. Line 113: I believe it is g/mL instead of mg/mL.

2. Line 126: I believe that the text in line 123-131 is the caption in Table 1. However, it appears to be part of the flowing text. Please correct it to avoid misunderstanding.

3. Line 142: The structure in Figure 1 is not necessary, as it is not alluded to in the rest of the text.

4. Line 216: Table 3 - It is necessary to make a more careful statistical analysis with the data. I suggest using the analysis methodology described by Lima et al., (2021) (10.1016/j.indcrop.2021.113282). The fact that the authors consider only the averages and not the standard deviations, makes them come to a wrong conclusion about the activity of the samples. This is evident from the fact that the control group was considered a class I repellent.

5. Line 219: It is necessary to quote the reference used for the classification of repellent agents. I believe it was McDonald et al., 1970 - Preliminary Evaluation of New Candidate Materials as Toxicants, Repellents, and Attractants Against Stored-product Insects.

6. Lines 235-257: This part of the article needs to be further discussed, as it is the central point of the work.
a) Authors should suggest hypotheses for essential oils to have such an ephemeral repellent action. Volatilization? Degradation? Then they should have a wide discussion, based on the literature, of the activity profile vs. time checked for ZEOP.
b) The results need to be compared with data from the literature for similar samples. Both free essential oils and in polymeric matrices.

Author Response

Reviewer 3

We are grateful to the Reviewer for their thoughtful and helpful comments on our manuscript. All their concerns have been addressed and changes in the text are highlighted in yellow. Please find below a point-by-point list of our answers 

Comments and Suggestions for Authors

I provided my comments as follows.

General comments:

The manuscript deals with the development of a essential oil-based bio-repellent against Triatomainfestans. The poly (ε-caprolactone) matrix loaded with Zuccagnia punctata essential oil was well characterized. However, the manuscript has important flaws and must be improved before being considered for publication in Molecules. 

The following are my considerations:

  1. Line 113: I believe it is g/mL instead of mg/mL.

Authors: We thank Reviewer #3 for her/his valuable comments. 

0.96 mg/mL is not correct, the value corresponds to 0.96 g/mL

  1. Line 126: I believe that the text in line 123-131 is the caption in Table 1. However, it appears to be part of the flowing text. Please correct it to avoid misunderstanding.

Authors: Exactly. A blank line was added to separate the caption and the text.

  1. Line 142: The structure in Figure 1 is not necessary, as it is not alluded to in the rest of the text.

Authors: Figure 1 was deleted as suggested 

  1. Line 216: Table 3 - It is necessary to make a more careful statistical analysis with the data. I suggest using the analysis methodology described by Lima et al., (2021) (10.1016/j.indcrop.2021.113282). The fact that the authors consider only the averages and not the standard deviations, makes them come to a wrong conclusion about the activity of the samples. This is evident from the fact that the control group was considered a class I repellent.

Authors: Repellency bioassays of and statistical analysis were carried out according to Talukder and Howse (1994). 

The presentation of the data in Tables 3 and 4 was modified. The values of the solvent control and the positive control of repellent activity were reviewed.

The repellent activity data were reviewed supported by additional experimental tests and statistical analyzes. Moreover, additional paragraphs referring to the repellent activity were incorporated in the text, accompanied by the respective bibliographic references.

  1. Line 219: It is necessary to quote the reference used for the classification of repellent agents. I believe it was McDonald et al., 1970 - Preliminary Evaluation of New Candidate Materials as Toxicants, Repellents, and Attractants Against Stored-product Insects.

Authors: The following reference that support the assay was included in section 4.2. Repellent activity against Triatoma infestans nymphs fifth instars: [38] Talukder, F. A.; Howse, P. E. Laboratory evaluation of toxic and repellent properties of the pithraj tree, Aphanamixis polystachya Wall & Parker, against Sitophilusoryzae (L.). Int. J. Pest Manag. 1994, 40, 274–279.

  1. Lines 235-257: This part of the article needs to be further discussed, as it is the central point of the work. a) Authors should suggest hypotheses for essential oils to have such an ephemeral repellent action. Volatilization? Degradation? Then they should have a wide discussion, based on the literature, of the activity profile vs. time checked for ZEOP. b) The results need to be compared with data from the literature for similar samples. Both free essential oils and in polymeric matrices.

Authors: This section was revised and improved as suggested.

Round 2

Reviewer 2 Report

Dear,

The authors' responses and corrections are satisfactory. I have no more comments.
I think the manuscript is publishable.

Sincerely,

Author Response

We are grateful to the Reviewer for their thoughtful and helpful comments on our manuscript. Please find below our answer 

English language and style are fine/minor spell check required

Response to Reviewer 2

English language and style were revised and corrected.

Reviewer 3 Report

Dear authors,

Congratulations on the good work done in correcting the manuscript. The text has been significantly improved. I have only one important consideration to make regarding the analysis of repellency results.

In tables 3 and 4, data from the control group indicate that it had an extremely negative repellency percentage (-100%) after 72 h. Thus, the control should be considered an attractor.
Please note that the behavior of the organisms tested with the control group was totally opposite to that of the group treated with DEET, which supports the conclusion that the control was strongly attractor and DEET was strongly repellent. Note also that the repellency percentage scale used in the article used as reference by the authors indicates that only samples with positive RP values ​​are considered repellent, while samples with values ​​close to zero are considered inactive (class 0).
Authors should redo the repellence test with the control group, as there is no reason to believe that there is repellent or attractant activity in this test.

Author Response

Reviewer 3

Comments and Suggestions for Authors:

Congratulations on the good work done in correcting the manuscript. The text has been significantly improved. I have only one important consideration to make regarding the analysis of repellency results.

In tables 3 and 4, data from the control group indicate that it had an extremely negative repellency percentage (-100%) after 72 h. Thus, the control should be considered an attractor.

Please note that the behavior of the organisms tested with the control group was totally opposite to that of the group treated with DEET, which supports the conclusion that the control was strongly attractor and DEET was strongly repellent. Note also that the repellency percentage scale used in the article used as reference by the authors indicates that only samples with positive RP values ​​are considered repellent, while samples with values ​​close to zero are considered inactive (class 0).

Authors should redo the repellence test with the control group, as there is no reason to believe that there is repellent or attractant activity in this test.

Response to Reviewer 3

We have received and appreciate the valuable opinion of the reviewer 3, and we would like to comment on some details that we consider relevant to our manuscript.

The blank control (acetone solvent) that the reviewer suggests with attractant activity is not possible, since in this control, UHPLC grade acetone and triatomines are used in the different stages, so that after the evaporation process there are no traces of any chemical substance that could exert a repellent or attractive effect on insects.

For the same reason, no grade is scored according to the scale used.

The insects in the target control have a random behavior, and on some occasions they can group together on one side of the Petri dish, without implying an attractive or repellent effect, since there would be no chemical compound that causes it. The control is for the purposes of live insect control.

The acetone blank control does not intervene in the calculation (formula) of the repellency value

The repellency test has been developed and statistically analyzed again in the first review, at the request of the reviewer, and the results demonstrate and support the repellent activity of the essential oil and the polymer containing the oil and also the reference compound DEET.

To avoid confusion for potential readers, a convenient solution that we will consider in the future in other reports is not to include this column in the results table. But we ask this time that you please consider keeping the respective column.

We hope our response is satisfactory for the final acceptance of our manuscript. However, we are willing to make changes if they suggest it again.